# Identification of novel myelodysplastic syndromes prognostic subgroups by integration of inflammation, cell-type composition, and immune signatures in the bone marrow

Sila Gerlevik[1†], Nogayhan Seymen[1†], Shan Hama[1†], Warisha Mumtaz[1], I Richard Thompson[1], Seyed R Jalili[1], Deniz E Kaya[1], Alfredo Iacoangeli[2,3,4,5], Andrea Pellagatti[6], Jacqueline Boultwood[6], Giorgio Napolitani[1], Ghulam J Mufti[1*], Mohammad M Karimi[1*]

[1]Comprehensive Cancer Centre, School of Cancer and Pharmaceutical Sciences, Faculty of Life Sciences & Medicine, King's College London, London, United Kingdom; [2]Department of Basic and Clinical Neuroscience, King's College London, London, United Kingdom; [3]Department of Biostatistics and Health Informatics, King's College London, London, United Kingdom; [4]NIHR BRC SLAM NHS Foundation Trust, London, United Kingdom; [5]Perron Institute for Neurological and Translational Science, University of Western Australia Medical School, Perth, Australia; [6]Nuffield Division of Clinical Laboratory Sciences, Radcliffe Department of Medicine, University of Oxford, Oxford, United Kingdom

*For correspondence:
ghulam.mufti@kcl.ac.uk (GJM);
mohammad.karimi@kcl.ac.uk
(MMK)

†These authors contributed
equally to this work

Competing interest: See page
14

Reviewing Editor: Satyajit
Rath, Indian Institute of Science
Education and Research (IISER),
India

**Abstract** Mutational profiles of myelodysplastic syndromes (MDS) have established that a relatively small number of genetic aberrations, including SF3B1 and SRSF2 spliceosome mutations, lead to specific phenotypes and prognostic subgrouping. We performed a multi-omics factor analysis (MOFA) on two published MDS cohorts of bone marrow mononuclear cells (BMMNCs) and CD34 + cells with three data modalities (clinical, genotype, and transcriptomics). Seven different views, including immune profile, inflammation/aging, retrotransposon (RTE) expression, and cell-type composition, were derived from these modalities to identify the latent factors with significant impact on MDS prognosis. SF3B1 was the only mutation among 13 mutations in the BMMNC cohort, indicating a significant association with high inflammation. This trend was also observed to a lesser extent in the CD34 + cohort. Interestingly, the MOFA factor representing the inflammation shows a good prognosis for MDS patients with high inflammation. In contrast, SRSF2 mutant cases show a granulocyte-monocyte progenitor (GMP) pattern and high levels of senescence, immunosenescence, and malignant myeloid cells, consistent with their poor prognosis. Furthermore, MOFA identified RTE expression as a risk factor for MDS. This work elucidates the efficacy of our integrative approach to assess the MDS risk that goes beyond all the scoring systems described thus far for MDS.

## eLife assessment

This manuscript uses public datasets of myelodysplastic syndrome (MDS) patients to undertake a multi-omics analysis of clinical, genomic, and transcriptomic datasets. **Useful** findings are provided by way of interesting correlations of specific mutations with inflammation and differing clinical

outcomes. The evidence is **solid** and interesting, and the manuscript is of substantive value to hematologists and clinical immunologists.

## Introduction

MDS are haematological diseases characterised by clonal proliferation due to genetic and epigenetic alterations within haematopoietic stem and progenitor cells (*Sperling et al., 2017*; *Mufti et al., 2008*). Thus far, the prediction of a patient's overall survival (OS) and event-free survival (EFS) has been predominantly dependent on the degree of peripheral blood cytopenias, bone marrow (BM) blast percentage, cytogenetics, and genetic features (*Zhang et al., 2022*; *Arber et al., 2016*; *Khoury et al., 2022*; *Greenberg et al., 2012*). This may, however, overlook other biological phenotypes and risk factors in MDS, including inflammation, age and aging characteristics, immune profile, BM cell-type composition, and expression of the noncoding sequences in particular retrotransposable elements (RTEs).

Inflammation can either suppress or promote cancer development (*Zhao et al., 2021*). Inflammation can be protective against malignancies, including MDS, through innate and adaptive immune responses, enhancing antitumor immunity by promoting the maturation and function of dendritic cells (DCs) and initiating effector T cell responses (*Ma et al., 2013*). However, low-level chronic inflammation can result in an immunosuppressive milieu preventing the innate and T-cell antitumor immunity (*McLaughlin et al., 2020*). Inflammaging is the process by which an age-related increase in chronic inflammation occurs, but the extent to which this process and other age-related events can impact the overall prognosis in MDS is yet to be uncovered (*Leonardi et al., 2018*; *Weeks et al., 2022*).

RTEs are genomic remnants of ancient DNA sequences that comprise a large part of the human non-coding genome and are evolutionarily silenced. RTEs include three classes: long terminal repeat (LTR) RTEs, and long and short interspersed nuclear elements, known as LINEs and SINEs, respectively. As part of the host defence mechanism, RTEs are silenced through DNA methylation and histone modifications in somatic cells (*Anwar et al., 2017*). Mutations in genes implicated in DNA methylation and histone modifications (DNMT3A, TET2, ASXL1, and IDH1/2) are frequently reported in MDS. It is postulated that global hypomethylation can reactivate RTEs in MDS cases with epigenetic mutations, particularly in DNMT3A mutant cases, as has previously been shown for other cancer types (*Wolff et al., 2010*; *Hur et al., 2014*). Despite the availability of multiple transcriptomics data for MDS (*Pellagatti et al., 2018*; *Pellagatti et al., 2010*; *Shiozawa et al., 2017*; *Choudhary et al., 2022*), a comprehensive assessment of RTE expression and its relationship to genetic variations or prognosis has not been documented.

Splicing factor (SF3B1 SRSF2, and U2AF1) mutations are the most common mutations in MDS (*Haferlach et al., 2014*; *Papaemmanuil et al., 2011*; *Papaemmanuil et al., 2013*; *Yoshida et al., 2011*). SF3B1 mutations are associated with the MDS ring sideroblasts type, good prognosis, and low leukemic transformation (*Papaemmanuil et al., 2011*). In contrast, SRSF2 mutations are associated with poor prognosis and are more prevalent in the male sex and older age (*Wu et al., 2016*). Recent studies have shown mutations in splicing factors can induce chronic innate immunity and enhance NF-κB signalling in MDS through aberrant splicing of various target genes. (*Choudhary et al., 2022*; *Lee et al., 2018*; *Smith et al., 2019*). These recent mechanistic studies have significantly advanced our understanding of the consequences of splicing factor mutation in human and model organisms. However, we still lack a systematic approach to integrate splicing factor mutations with other players in the tumour microenvironment, including immune profile and BM cell-type composition.

Recent work has explored the relationship between transcriptional signatures and critical signalling pathways to determine survival prognosis and diagnostic efficacy in MDS patient cohorts (*Tuerxun et al., 2022*). However, no studies to date integrated clinical MDS phenotypes, RTE expression, cell-type composition, and immune and aging gene signatures. This study employs MOFA for a comprehensive analysis of three data modalities (clinical, genotypic, and transcriptomic) and seven different 'views' derived from these modalities to identify the factors that may impact MDS prognosis. MOFA could not identify any factor representing splicing factor mutations; hence, we examined our entire feature sets from cell-type composition, immune profile, and inflammation/aging views to identify the features associated with mutations in SF3B1 and SRSF2 genes in MDS cohorts.

## Results

## MOFA identified latent factors linking different views in multimodal MDS data

We utilised two RNA-seq datasets for MDS (*Supplementary file 1*). The first was data from BMMNCs of 94 MDS patients obtained from the *Shiozawa et al., 2017* that is enriched with splicing factor mutations (*Supplementary file 2*). The second dataset utilised bone marrow CD34 + haematopoietic stem and progenitor cells (HSPCs) data that *Pellagatti et al., 2018* derived from 82 MDS patients, which again focused on MDS cases with splicing factor mutations (*Supplementary file 2*).

We applied MOFA to identify latent factors within BMMNC and CD34 + MDS cohorts (*Figure 1a*). To run MOFA on these two MDS cohorts, three (immune profile, inflammation/aging profile, and cell-type composition) out of seven views were derived from RNA-seq by applying *singscore* (*Foroutan et al., 2018*) on RNA-seq gene expression. Each of these three views were carried forward in the workflow by a number of gene sets, and per gene set, *singscore* generated relative gene signature scores for all samples within each cohort (*Figure 1a* and *Supplementary file 3*). The other four views were clinical numeric, clinical categorical, genotype, and RTE expression.

We have provided the data generated for these seven views to MOFA in two separate runs for BMMNC and CD34 + MDS cohorts. For each cohort, we could identify ten factors (minimum explained variance 2% in at least one biological view) from the 15 default factors generated by MOFA (*Figure 1b, c*). These factors displayed a relationship with different biological views, with Factor 1 as the most dominant factor, linking immune profile, cell-type composition, and inflammation/aging profile in both cohorts. Further to this, in the BMMNC cohort, a high level of variance for RTE expression was explained by Factor 1 (*Figure 1b*). We also observed similar trends in terms of the level of variance explained by the identified factors for the biological views in both cohorts, with the only notable difference being a greater level of variance explained for genotype data and a lower level of variance explained for RTE expression in the BM CD34 + cohort versus the BMMNC cohort (*Figure 1d, e*). Additionally, dividing the patients based on low (first quartile), intermediate (second and third quartile), and high (fourth quartile) levels of Factor 1 in both cohorts could successfully stratify patients in the dimensionality reduction plots obtained from applying principal component analysis (PCA) on gene expression data from both cohorts (*Figure 2—figure supplement 1*).

MOFA also identified the highly weighted features within each factor in the BMMNC and BM CD34 + cohorts (*Figure 2a, b*), in which each feature belongs to a specific biological view. We further characterised Factor 1 as the most dominant factor linking multiple features from different views. We observed that high Factor 1 in the BMMNC cohort represented patients who have cells with stem and progenitor-like characteristics. This includes but is not limited to a positive correlation with features comprising progenitor-like and HSC-like (*Figure 2c*). In contrast, there is an inverse correlation between GMP/GMP-like and Factor 1 scores (*Figure 2—figure supplement 2c*). SINE: Alu expression is increased in patients high in Factor 1 within this cohort, representing a group that may have significant levels of genetic instability (*Figure 2—figure supplement 2a*). Moreover, Factor 1 correlates with the following immunology features: increased T-helper 1 (Th1) cells, and a decrease in certain immune cells, especially in neutrophils and exhausted CD8 + T cells (*Figure 2—figure supplement 2b*). The relationship between Th1 and SINE elements is particularly interesting given the role of Th1 cell activation in host defence against inflammation that may be modulated by SINE activation. Furthermore, there is a modest correlation between immunosenescence and exhausted CD8 + T cell scores, particularly in patients with higher levels of GMPs, demonstrating the ineffectiveness of the immune system in GMP-dominant patients (*Figure 2c*).

MOFA analysis of the CD34 + MDS Cohort revealed factors associated with different immune signatures. Factor 1 is associated with signatures of differentiated myeloid cells (*Figure 2d* and *Figure 2—figure supplement 3c*), Factor 2 with HSPC, and Factor 4 with GMP progenitor cells, suggesting that these factors might correlate with distinct differentiation states of MDS blasts (*Figure 2b*). Factor 1 is also associated with cytolytic and cytotoxicity scores (*Figure 2d* and *Figure 2—figure supplement 3b*). Since CD34 + cells used for this analysis were purified using anti-CD34 conjugated beads and not FACS sorted, this signal might depend on the presence of T or NK cell contaminations in the CD34 + fraction. The trend that we observed for Factor 1 prompted us to conduct a differential expression and gene set enrichment analysis (GSEA) to identify the pathways that are dysregulated in patients having high (fourth quartile) versus low (first quartile) levels of Factor 1. Interestingly, we found upregulation

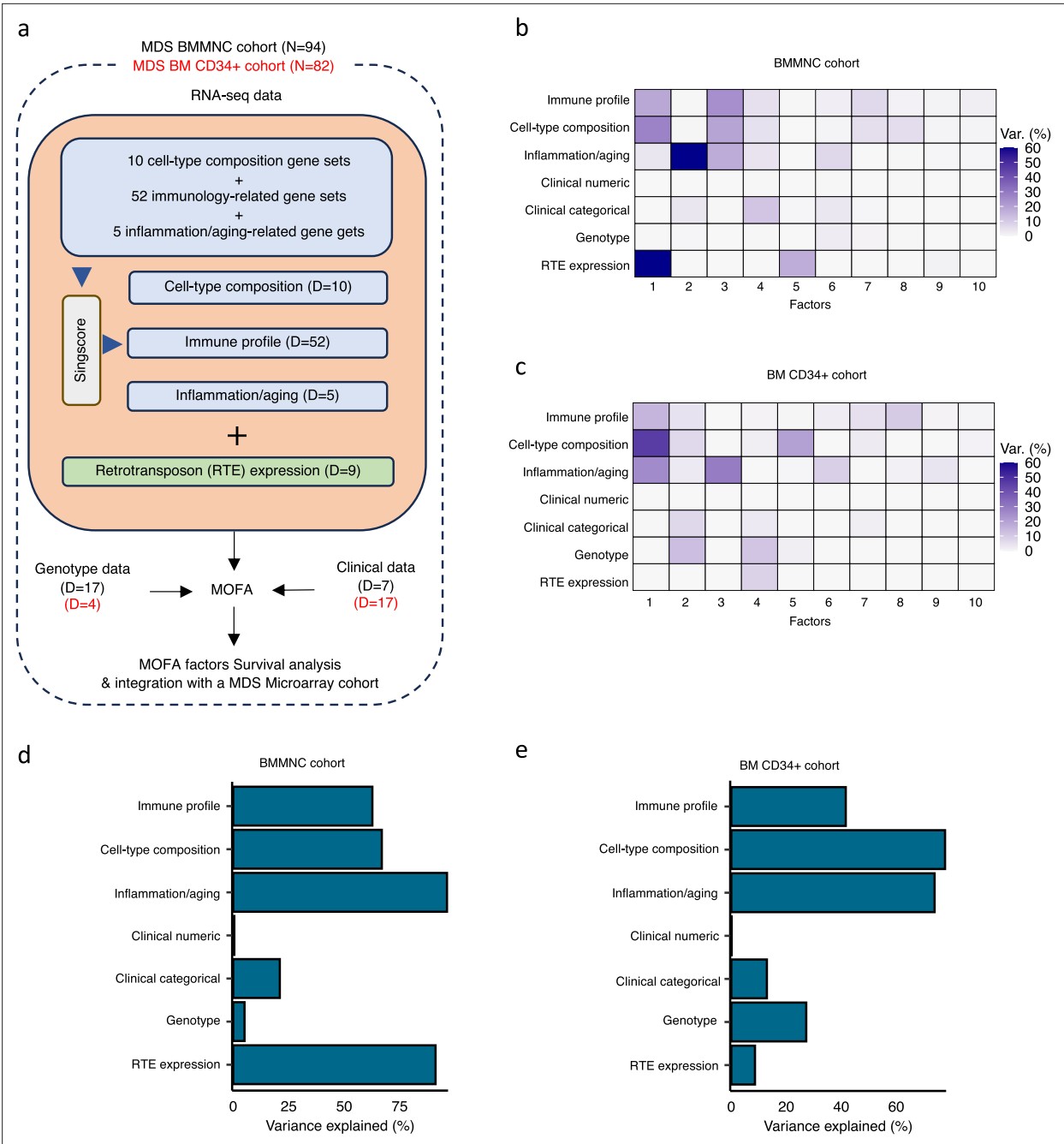

**Figure 1.** Schematic of multi-omics factor analysis (MOFA) workflow overview, downstream analyses, and factor determination in the bone marrow mononuclear cell (BMMNC) and bone marrow (BM) CD34 + cohorts. (**a**) RNA-seq, genotype, and clinical data were obtained from BMMNC samples of 94 myelodysplastic syndromes (MDS) patients from Shiozawa et al. and BM CD34 + samples of 82 patients from Pellagatti et al. studies. We generated seven views of the data where three of which were derived from RNA-seq data after applying Singscore: immune profile, cell-type composition, and inflammation/aging. The other four views were clinical numeric and categorical (***Supplementary file 4***), genotype, and retrotransposable element (RTE) expression. The data were put through MOFA to identify latent factors and the variance decomposition by factors. The number of features (dimensions) per view is abbreviated by 'D.' (**b, c**) The determined factors for the BMMNC and BM CD34 + cohorts and the percentage of explained variance for each view per identified factor were shown. (**d, e**) Bar charts depict the total variance explained for each biological data view by all the factors combined in the BMMNC and BM CD34 + cohorts.

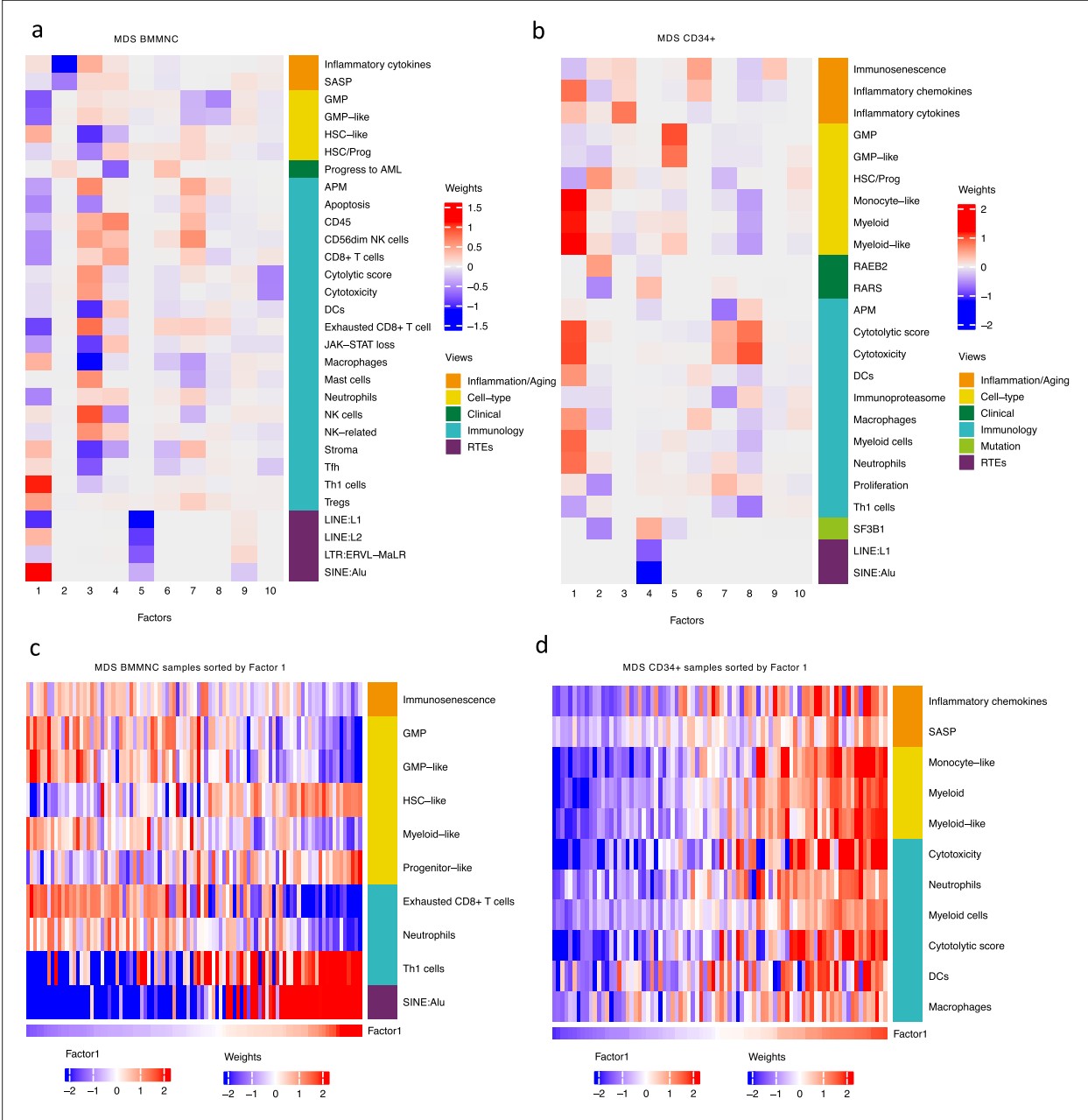

**Figure 2.** Breakdown of important features for each factor generated by multi-omics factor analysis (MOFA). (**a, b**) The important features with high weights for each biological view per factor were shown for the BMMNC and bone marrow (BM) CD34+. Blue represents features with the inverse correlation with the factor, and red shows the positive correlation. (**c, d**) Characterisation of Factor 1 in the BMMNC and BM CD34 + cohorts, showing only those features highly influencing Factor 1 for the patients in these cohorts. Patients were sorted by Factor 1 values.

The online version of this article includes the following figure supplement(s) for figure 2:

**Figure supplement 1.** Factor 1 from the bone marrow mononuclear cell (BMMNC) and CD34 + myelodysplastic syndromes (MDS) cohorts stratified the patients in the gene expression principal component analysis (PCA) plots.

**Figure supplement 2.** The top features influencing Factor 1 form various biological views in the bone marrow mononuclear cell (BMMNC) cohort.

**Figure supplement 3.** The top features influencing Factor 1 form various biological views in the CD34 + cohort.

**Figure supplement 4.** Gene set enrichment analysis (GSEA) identified upregulation of inflammation related pathways in patients having high (fourth quartile) versus low (first quartile) levels of Factor 1 in the CD34 + cohort.

**Figure supplement 5.** Characterisation of Factor 4 in the bone marrow mononuclear cell (BMMNC) cohort using gene signatures from multiple biological views.

**Table 1.** Association of Factors in the bone marrow mononuclear cell (BMMNC) cohort to overall survival and event-free survival. Univariate and multivariate (controlling for sex and age) Cox regression were undertaken for patients in the BMMNC cohort to determine the relationship between the identified multi-omics factor analysis (MOFA) factors and overall survival (OS)/event-free survival (EFS). The regression was done using 95% confidence intervals, with the outcome shown as hazard ratios (HR). The table depicts P-values for both univariate and multivariate analyses for each Factor. Statistical significance was achieved with Factors 2, 4, and 9 for univariate and/or multivariate Cox regression analyses.

| | Overall survival | | | | Event-free survival | | | |
|---|---|---|---|---|---|---|---|---|
| | Univariate | | Multivariate | | Univariate | | Multivariate | |
| Factors | Hazard ratio (CI) | p-value | Hazard ratio (CI) | p-value | Hazard ratio (CI) | p-value | Hazard ratio (CI) | p-value |
| Factor 1 | 0.74 (0.47–1.18) | 0.209 | 0.36 (0.16–0.81) | 0.013* | 0.76 (0.48–1.20) | 0.242 | 0.69 (0.34–1.38) | 0.294 |
| Factor 2 | 1.55 (0.95–2.53) | 0.082 | 1.64 (0.71–3.78) | 0.248 | 1.70 (1.04–2.77) | 0.033* | 2.27 (1.07–4.81) | 0.033* |
| Factor 3 | 0.56 (0.22–1.41) | 0.218 | 0.33 (0.07–1.54) | 0.157 | 0.47 (0.18–1.22) | 0.121 | 0.28 (0.06–1.34) | 0.11 |
| Factor 4 | 0.41 (0.26–0.65) | <0.001* | 0.39 (0.17–0.90) | 0.028* | 0.48 (0.34–0.66) | <0.001* | 0.66 (0.36–1.20) | 0.176 |
| Factor 5 | 0.76 (0.52–1.10) | 0.148 | 0.68 (0.40–1.15) | 0.148 | 0.80 (0.55–1.17) | 0.250 | 0.84 (0.49–1.44) | 0.527 |
| Factor 6 | 1.27 (0.85–1.91) | 0.238 | 1.26 (0.71–2.23) | 0.431 | 1.28 (0.85–1.93) | 0.230 | 1.21 (0.72–2.02) | 0.478 |
| Factor 7 | 1.05 (0.66–1.66) | 0.832 | 0.76 (0.40–1.44) | 0.4 | 0.98 (0.62–1.55) | 0.943 | 0.96 (0.56–1.64) | 0.879 |
| Factor 8 | 1.73 (1.06–2.82) | 0.029* | 1.08 (0.57–2.02) | 0.818 | 1.73 (1.07–2.79) | 0.025* | 1.23 (0.62–2.43) | 0.554 |
| Factor 9 | 0.56 (0.38–0.84) | 0.005* | 0.49 (0.28–0.85) | 0.012* | 0.59 (0.40–0.87) | 0.007* | 0.45 (0.26–0.80) | 0.007* |
| Factor 10 | 1.07 (0.69–1.67) | 0.757 | 0.76 (0.43–1.34) | 0.336 | 1.07 (0.69–1.66) | 0.755 | 0.76 (0.42–1.38) | 0.371 |

of inflammatory, Interferons, TNFA, and JAK-STAT signalling pathways within cancer hallmark gene sets (*Figure 2—figure supplement 4a*) and upregulation of chemokines and Neutrophils amongst Reactome gene sets (*Figure 2—figure supplement 4b*) supporting the trend observed by MOFA (*Figure 2d* and *Figure 2—figure supplement 3b*). This analysis suggests Factor 1 as an immune-active factor associated with a high-level of cytotoxicity and inflammation in the CD34 + MDS Cohort.

## The Cox regression models identified the high expression of RTEs as a risk factor and inflammation as a protective factor in MDS

We explored the power of the latent factors generated by MOFA as predictors of MDS survival. Univariate and multivariate Cox regression models identified 3 of the 10 factors identified by MOFA were significantly associated with OS or EFS in the BMMNC cohort (*Table 1*). The univariate analysis was conducted to investigate the association of the factors with survival, and the multivariate analysis controlled for sex and age. Our univariate analysis demonstrated that Factors 4 and 9 significantly impact OS (HR: 0.39; p=0.028 and HR: 0.49; p=0.012, respectively) and EFS (HR: 0.48; p<0.001 and HR: 0.59; p=0.007, respectively) in patients in the BMMNC cohort (*Table 1*). Interestingly, the impact of Factor 4 on EFS became insignificant after controlling for sex and age. Though no significance was associated with OS and Factor 2, EFS in both the univariate and multivariate analyses exhibited a statistically significant association with this factor (HR: 1.7; p=0.033 and HR: 2.27; p=0.033, respectively) (*Table 1*). The 10 factors identified by MOFA in the BM CD34 + cohort did not show any significance associated with MDS overall survival (*Supplementary file 5*).

Dividing the patients based on low (first quartile), intermediate (second and third quartile), and high (fourth quartile) levels of Factors 2, 4, and 9 in the BMMNC cohort, we found Factors 4 and 9 exert a protective influence over the prognosis of MDS patients (*Figure 2—figure supplement 5a* and *Figure 3a*, respectively), whilst higher levels of Factor 2 predict a poorer prognosis for these same patients (*Figure 4a*). Upon investigation of the RTE absolute loading of features in Factor 9, patients with low Factor 9 levels showed increased LTR: ERV1, SINE: MIR and SINE: Alu demonstrating the high level of RTE expression as a potential risk factor for MDS (*Figure 3a–b*). In contrast, for Factor 2, some component weights for inflammation/aging including inflammatory cytokines and SASP were

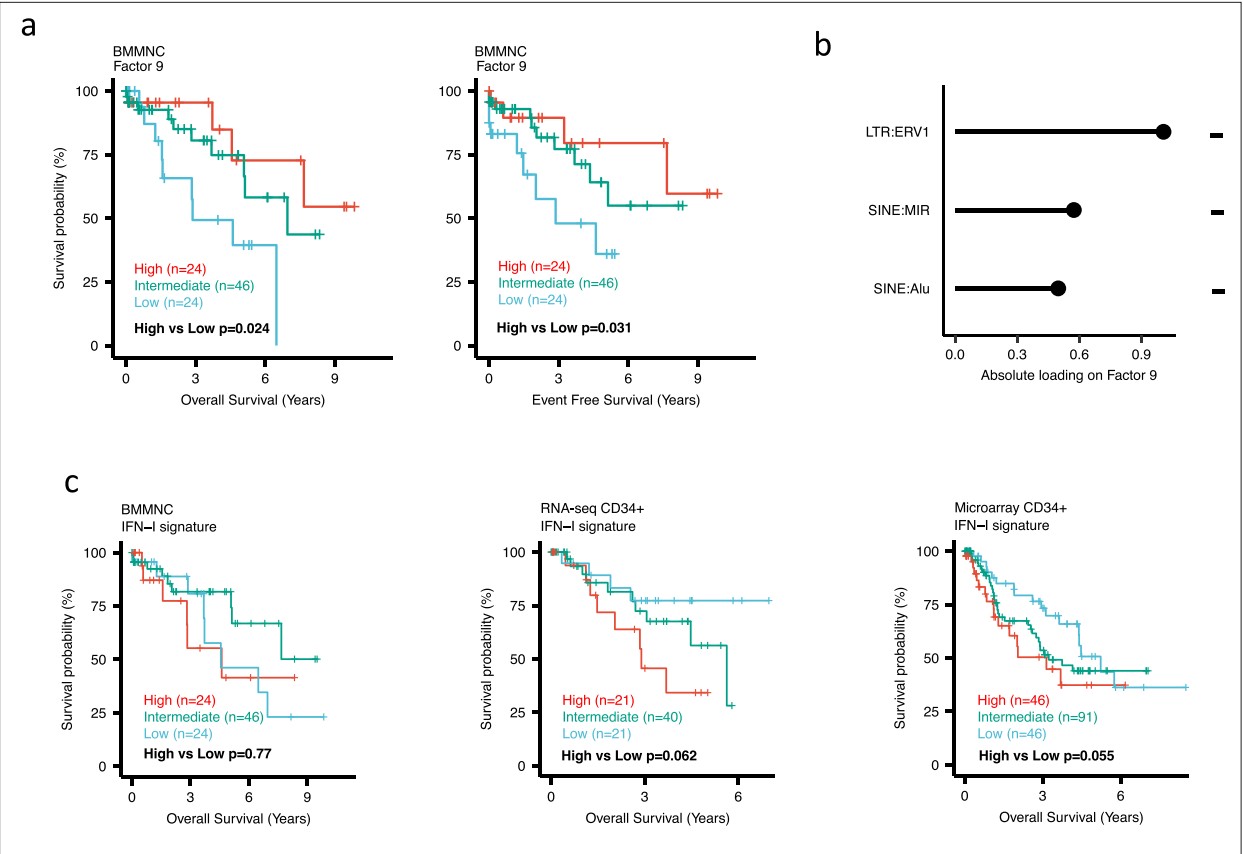

**Figure 3.** Impact of Factors 9 and IFN-I levels on myelodysplastic syndromes (MDS) prognosis. (**a**) Kaplan-Meier plots for the bone marrow mononuclear cell (BMMNC) cohort where patients were split based on low (first quartile), intermediate (second and third quartile), and high (fourth quartile) levels of Factor 9. (**b**) The absolute loading of the top three features affecting Factor 9 in retrotransposable element (RTE) expression view in the BMMNC cohort. (**c**) Kaplan-Meier plots where patients were split into quartiles (high 25%, low 25%, and intermediate 50%) for survival analyses depending on their IFN-I signature score levels in the BMMNC, RNA-seq CD34+, and Microarray CD34 + cohorts. All p-values were calculated using a log-rank test on overall and event-free survival values of high versus low groups.

increased once the Factor 2 level was decreased suggesting the secretion of cytokines and the downstream inflammation as a protective factor for MDS (**Figure 4a–b**).

Factor 4 in the BMMNC cohort shows several data views and displays a more specific phenotype of the patients. This phenotype includes the accumulation of immune cells, such as CD56dim NK cells, CD8 + T cells, DCs, and neutrophils (**Figure 2a**). Thus, Factor 4 points to a more immune-active disease. In line with this, there is a decreased rate of the 'progress to AML' phenotype (**Figure 2a** and **Figure 2—figure supplement 5b**) in patients with high levels of Factor 4. In contrast, for the patients showing low levels of Factor 4, we observed that the number of healthy stem and progenitor cells (HSCs) decreased (**Figure 2—figure supplement 5c**), whereas the number of malignant HSCs (HSC-like) increased (**Figure 2—figure supplement 5c, d**). Most notably, the BM of these patients has a high content of stroma and a low content of leukocytes, represented by a low CD45 marker score (**Figure 2—figure supplement 5c and e**). Overall, this result suggests that for a subset of MDS patients, represented by the low level of Factor 4, the haematopoiesis is impaired due to the depletion of healthy HSCs, leading to the decrease in the number of leukocytes, an increase of stroma, and features of secondary AML.

## BM CD34+ cohorts support a better prognosis for high-inflamed cases but a poor prognosis for IFN-1-induced cases

To better understand the drivers of prognosis for MDS, patients were split into quartiles (high 25%, low 25%, and intermediate 50%) for survival analyses depending on their levels for individual features (inflammatory cytokines, inflammatory chemokines, and IFN-I signature) in the two MDS cohorts. In

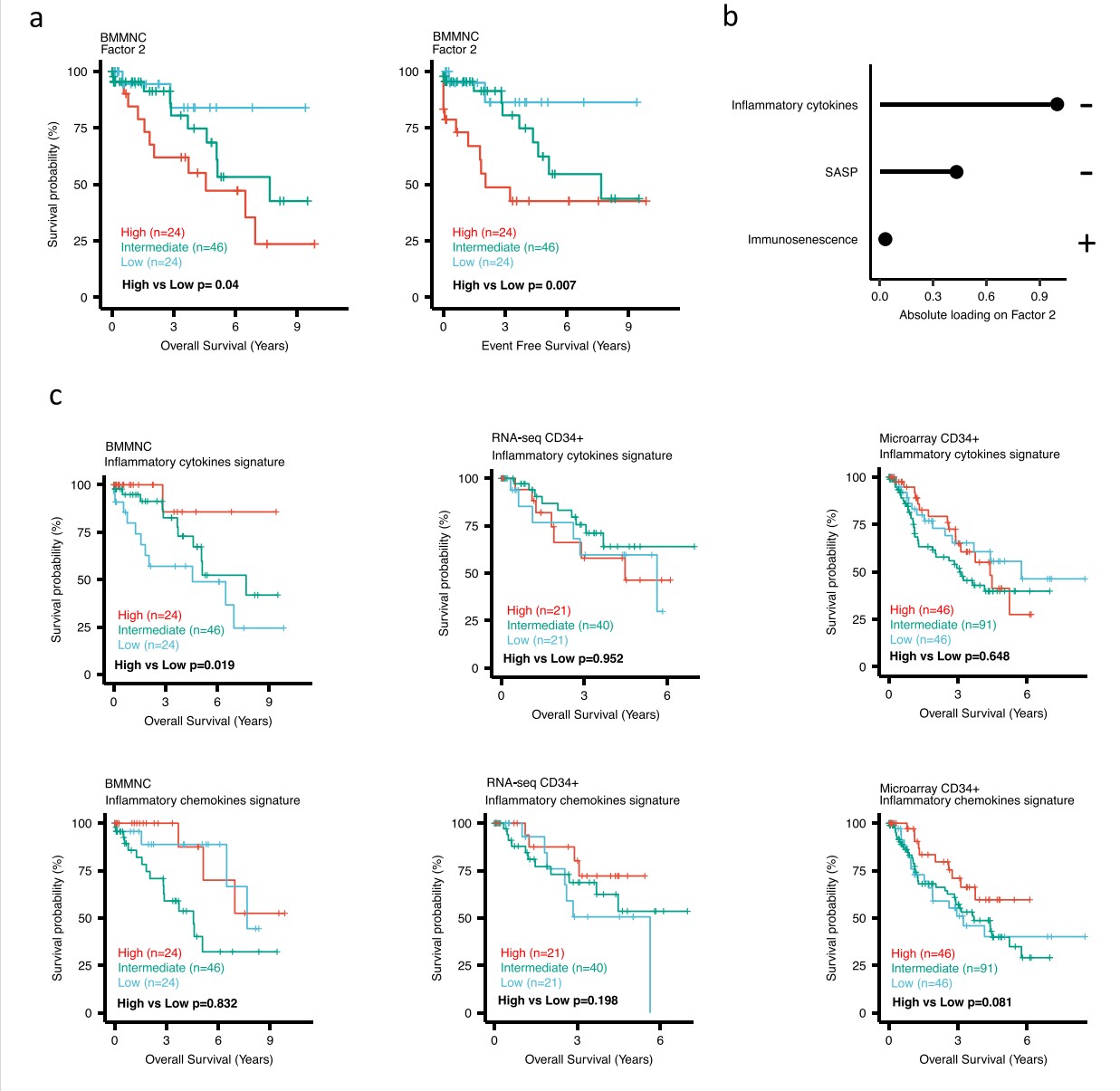

**Figure 4.** Impact of Factors 2 and inflammation levels on myelodysplastic syndromes (MDS) prognosis. (**a**) Kaplan-Meier plots for the bone marrow mononuclear cell (BMMNC) cohort where patients were split based on low (first quartile), intermediate (second and third quartile), and high (fourth quartile) levels of Factor 2. (**b**) The absolute loading of the top three features affecting Factor 2 in inflammation/aging biological view in the BMMNC cohort. (**c**) Kaplan-Meier plots where patients were split into quartiles (high 25%, low 25%, and intermediate 50%) for survival analyses depending on their inflammatory cytokines and chemokines levels in the BMMNC, RNA-seq CD34+, and Microarray CD34 + cohorts. All p-values were calculated using a log-rank test on overall and event-free survival values of high versus low groups.

addition, a microarray gene expression dataset obtained from the HSPCs isolated from BM of 183 MDS patients (*Pellagatti et al., 2010*) was also included in this analysis. Recall, in the BMMNC cohort, Factor 2 was primarily represented by the inflammation/aging biological view (*Figure 4b*) and Factor 9 was represented by RTE expression (*Figure 3b*). To confirm the isolated impact of the aforementioned features on patient survival in CD34 + cohorts, we created survival plots for each of the three cohorts separating patients into quartiles based on high, low, and intermediate levels of the features. The only situation in which we observed statistical significance was the high score of inflammatory cytokines in the BMMNC cohort, which was associated with superior OS (p=0.019) (*Figure 4c*) supporting the strong contribution of this feature to Factor 2 (*Figure 4b*). For the rest of the analyses, statistical significance was not achieved, though there may be a trend pertaining to higher inflammatory chemokines

signature and better prognosis in CD34 + cohorts (*Figure 4c*). Further to this, a higher IFN-I signature score was also related to a poorer prognosis in CD34 + cohorts (*Figure 3c*). Strikingly, this pattern was not observed in the BMMNC cohort (*Figure 3c*), suggesting an alternative mechanism other than IFN-I activation can contribute to poor MDS prognosis due to retroviral activations (*Figure 3a–b*).

## SF3B1 mutant cases show low HSPC content and high levels of inflammation in the BMMNC and BM CD34 + cohorts

When investigating the inflammation/aging and cell-type views in both the BMMNC and BM CD34 + cohorts, we identified a significant association between the occurrence of SF3B1 mutation and the high level of inflammatory chemokines (BMMNC p<0.01; BM CD34 + p<0.05) (*Figure 5a–c*). However, we needed to know whether the co-occurrence of other mutations with SF3B1, as confounding factors, can affect the statistical significance of the association of SF3B1 mutation with inflammatory chemokines. Therefore, we applied a multiple linear regression model to examine the association between various mutations (covariates) and inflammatory chemokines while controlling the effect of other mutations. Strikingly, out of 13 different mutations in the BBMNC cohort, the SF3B1 mutation was the only mutation showing a significant association with the inflammatory chemokine level in the multiple linear regression model (p<0.0001) (*Table 2*).

In both cohorts, the SF3B1 mutant pertained to decreased HSC/Prog cells (BMMNC p<0.05, BM CD34 + p<0.01) (*Figure 5a–b*). Furthermore, the BMMNC cohort showed a statistically significant increase in myeloid, HSC-like, and promo-like scores, along with inflammatory cytokines level (p<0.05) in SF3B1 mutant cases (*Figure 5a–b*). Applying multiple linear regression models and controlling the effect of other mutations supported the identified correlation between SF3B1 mutation and inflammatory cytokines level (p<0.02) (*Table 2*). We then examined the Immunology view in the BMMNC cohort to see whether we could identify any immune signature that the SF3B1 mutation may influence (*Figure 5—figure supplement 1a*). We found that SF3B1 mutants tend to have low levels of leukocytes but high levels of stroma and macrophages (*Figure 5d*). This result supports previous findings by *Pollyea et al., 2021*; *Pollyea et al., 2019* demonstrating the generation of inflammatory cytokines due to macrophage activation in human BM samples obtained from SF3B1 mutant patients.

We also divided the SF3B1 mutant cases from the BMMNC cohort into high versus low groups depending on their levels of inflammatory chemokines and cytokines. Although survival plots did not reach statistical significance, the trend shows that SF3B1 mutant cases with higher inflammation tend to survive better than wild-type SF3B1 (*Figure 5f–g*). This trend indicates that the SF3B1 mutation perhaps only induces macrophage activation to survive better in a subset of mutant cases.

## SRSF2 mutant cases show high GMP content and high levels of senescence and immunosenescence

When assessing inflammation/aging and cell-type features for MDS SRSF2 mutant cases in both cohorts, we saw a statistically significant increase in GMP and GMP-like features (p<0.05), which are precursor cells to granulocytes and monocytes (*Figure 6a–c*). The increased frequency of GMPs is an inherent feature of high-risk MDS (*Pang et al., 2013*; *Will et al., 2012*). In the BMMNC cohort, within *SRSF2* mutant cases, there was a statistically significant decrease in healthy myeloid features (p<0.05), along with an increase in myeloid-like (malignant myeloid) and senescence/immunosenescence (p<0.05) (*Figure 6a–b*). Further analyses of the BMMNC cohort indicated that SRSF2 mutants tend to have low levels of T cells and a reduction of T cell activity. The levels of several immune cells were decreased in SRSF2 mutants, including T, Th1, and Treg cells (p<0.05) (*Figure 6d* and *Figure 5—figure supplement 1b*). Patients with SRSF2 mutation, related to poorer prognosis in literature, also had a lower cytolytic score (p<0.001) and increased central memory cells (p<0.01) (*Figure 6d*). Taken together, this result finds relevance to the immunosenescence and the reduction of the T cells and their cytolytic activity in SRSF2 mutant MDS, leading to a severe and high-risk situation for MDS.

## Discussion

We applied MOFA to seven biological views derived from two MDS patient cohorts. MOFA could identify latent and important phenotypes from multimodal MDS data. Notably, we identified RTE expression as a risk factor and inflammation as a protective factor in MDS. Moreover, we uncovered

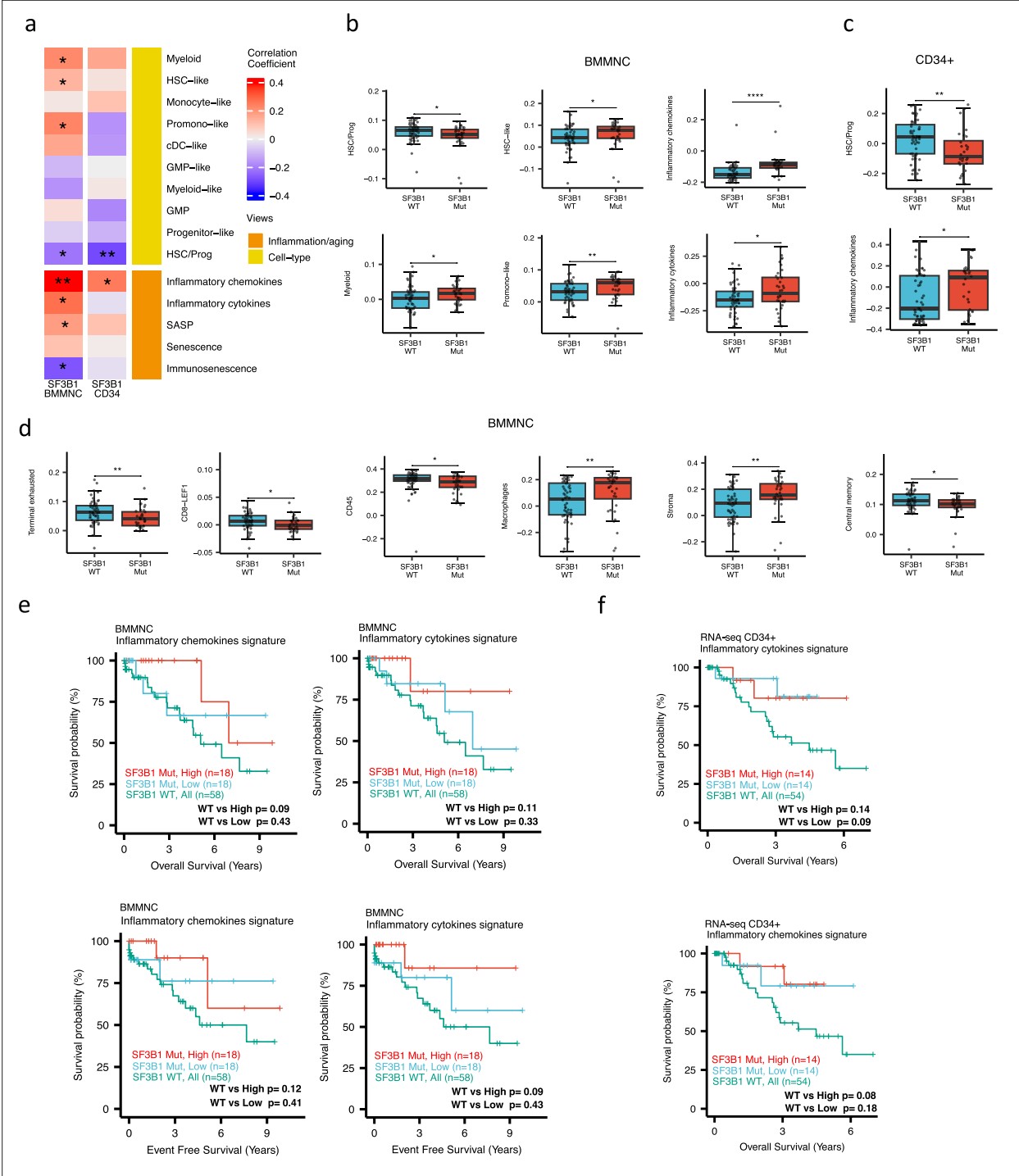

**Figure 5.** Characterisation of SF3B1 mutant myelodysplastic syndrome (MDS) using gene signatures from multiple biological views. (**a**) Association of a subset of inflammation/aging and cell-type features with SF3B1 mutation in the BMMNC and bone marrow (BM) CD34 + cohorts, with red depicting a positive correlation and blue an inverse correlation with SF3B1 mutation. The significances were calculated with the Wilcox rank-sum test, and the significant associations were shown by * (p<0.05), ** (p<0.01), or *** (p<0.001). (**b**, **c**) Boxplots comparing the levels of the significant individual features from the cell type and inflammation/aging biological views for SF3B1 mutant versus SF3B1 wild-type (WT) cases in the BMMNC (nMut = 36 vs. nWT = 58) and CD34+ (nMut = 28 vs. nWT = 54) cohorts. (**d**) Boxplots comparing the levels of the significant individual features from the immune profile biological views for SF3B1 mutant versus SF3B1 WT cases in the BMMNC cohort. (**e, f**) Kaplan-Meier plots displaying overall and event-free survivals for SF3B1 mutant cases split by high and low levels of inflammatory cytokines and chemokines versus SF3B1 mutant WT in the BMMNC and CD34+

*Figure 5 continued on next page*

*Figure 5 continued*

cohorts. Event-free survival data was only available for the BMMNC cohort. All p-values were calculated using a log-rank test on overall and event-free survival values of SF3B1 mutant high and low versus wild-type (WT) groups.

The online version of this article includes the following figure supplement(s) for figure 5:

**Figure supplement 1.** Correlation of selected immune profile gene sets with SF3B1 and SRSF2 mutations in the myelodysplastic syndromes (MDS) bone marrow mononuclear cell (BMMNC) cohort.

that Factor 4 correlated with progress to AML, with patients lower in this factor more likely to develop AML. Low level of Factor 4 is related to a phenotype with low frequency of leukocytes; patients also typically had increased malignant HSCs and stroma. Our analysis shows that these patients probably have a defect in haematopoiesis, preventing the production of sufficient blood cells and ultimately allowing for the stroma to invade the whole marrow (*Tripodo et al., 2011*; *Vega et al., 2002*). Although this observation is not novel, it is confirmatory of prior researches and demonstrates the power of our approach for integrating multimodal MDS data.

Literature has clearly defined the relationship between the SF3B1 mutation and long OS and EFS, with a low risk of progression to AML (*Wu et al., 2016*; *Migdady et al., 2018*; *Tang et al., 2019*). We showed that SF3B1 mutant MDS cases tend to have high levels of inflammation, perhaps due to macrophage activation, and thus confers a good prognosis for patients in terms of anti-tumour activity. Therefore, inflammation might help MDS survival. Further division of SF3B1 mutant patients into high and low levels of inflammation shows that the SF3B1 mutants with higher inflammation can generally survive better despite not being statistically significant. Investigating the synergic factors including epigenetic factors remains elusive. Enriching the multi-omics data with new modalities, including epigenomes, CyTOF, and Luminex cytokine/chemokine data in the future might help us understand why SF3B1 mutants show distinct patterns in terms of inflammation and survival.

Our work revealed that SRSF2 mutant MDS cases show a reduction of T cells. The decrease in the ability of patients to accumulate T cells, such as Th1 cells, which play crucial roles in modulating the

**Table 2.** Multiple linear regression analysis to study the association between various mutations (covariates) and inflammatory chemokines and cytokines.

SF3B1 mutation was the only mutation showing a significant association with the inflammatory chemokines level in the multiple linear regression model (p<0.001). The association between SF3B1 mutation and the inflammatory cytokines is also the most significant association among other mutations.

| Genes | Inflammatory chemokines | | | | Inflammatory Cytokines | | | |
|---|---|---|---|---|---|---|---|---|
| | Estimates | Std. error | t-value | p-value | Estimate | Std. error | t-value | p-value |
| ASXL1 (*n=21*) | –0.001 | 0.019 | –0.036 | 0.972 | –0.052 | 0.039 | –1.319 | 0.191 |
| CBL (*n=8*) | –0.02 | 0.029 | –0.69 | 0.492 | –0.008 | 0.061 | –0.13 | 0.897 |
| CUX1 (*n=9*) | –0.006 | 0.027 | –0.218 | 0.828 | –0.091 | 0.057 | –1.596 | 0.115 |
| DNMT3A (*n=7*) | –0.042 | 0.029 | –1.437 | 0.155 | 0.025 | 0.062 | 0.401 | 0.69 |
| EZH2 (*n=6*) | 0.024 | 0.031 | 0.776 | 0.44 | –0.13 | 0.066 | –1.967 | 0.053 |
| IDH1 (*n=5*) | 0.017 | 0.038 | 0.453 | 0.652 | –0.01 | 0.081 | –0.128 | 0.898 |
| JAK2 (n=5) | –0.003 | 0.032 | –0.107 | 0.915 | 0 | 0.068 | –0.004 | 0.997 |
| RUNX1 (*n=7*) | –0.006 | 0.03 | –0.209 | 0.835 | –0.003 | 0.064 | –0.045 | 0.964 |
| SF3B1 (*n=36*) | 0.059 | 0.016 | 3.708 | <0.001*** | 0.081 | 0.034 | 2.38 | 0.020* |
| SRSF2 (*n=17*) | 0 | 0.025 | –0.009 | 0.993 | –0.044 | 0.054 | –0.814 | 0.418 |
| STAG2 (*n=5*) | –0.039 | 0.035 | –1.123 | 0.265 | –0.159 | 0.074 | –2.153 | 0.035* |
| TET2 (*n=21*) | 0.031 | 0.02 | 1.523 | 0.132 | 0.013 | 0.043 | 0.304 | 0.762 |
| U2AF1 (*n=9*) | 0.056 | 0.028 | 1.97 | 0.053 | 0.153 | 0.06 | 2.54 | 0.013* |

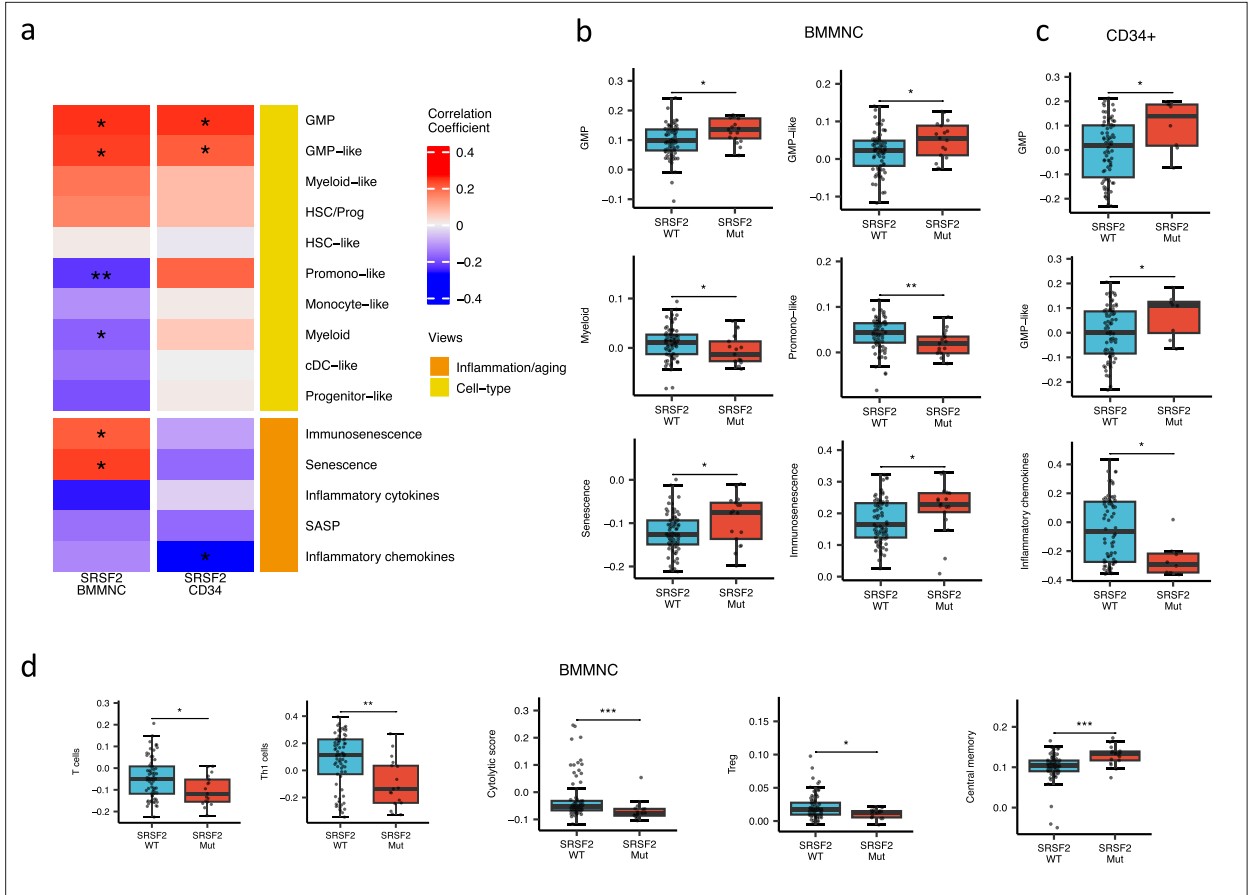

**Figure 6.** SRSF2 mutant myelodysplastic syndrome (MDS) is catheterised by high GMP content and high levels of senescence and immunosenescence. (**a**) Association of a subset of inflammation/aging and cell-type features with SRSF2 mutation in the bone marrow mononuclear cell (BMMNC) and bone marrow (BM) CD34 + cohorts, with red depicting a positive correlation and blue an inverse correlation with SRSF2 mutation. The significances were calculated with the Wilcox rank-sum test, and the significant associations were shown by * (p<0.05), ** (p<0.01), or *** (p<0.001). (**b–c**) Boxplots comparing the levels of the significant individual features from the cell-type and inflammation/aging biological views for SRSF2 mutant versus SRSF2 wild-type (WT) cases in the BMMNC (nMut = 17 vs. nWT = 77) and CD34+ (nMut = 8 vs. nWT = 74) cohorts, respectively. (**d**) Boxplots comparing the levels of the significant individual features from the immune profile biological views for SRSF2 mutant versus SRSF2 WT cases in the BMMNC cohort.

The online version of this article includes the following figure supplement(s) for figure 6:

**Figure supplement 1.** Positive correlation between programmed death-ligand 1 (PD-L1) expression and senescence score in bone marrow (BM) CD34 + RNA seq cohort.

killing of tumour cells may cause worse outcomes for SRSF2 mutant patients (**Knutson and Disis, 2005**).

We also observed increased immunosenescence levels in SRSF2 mutants. Recent studies have shown that the expression of programmed death-ligand 1 (PD-L1) protein is significantly elevated in senescent cells (**Wang et al., 2022**; **Pippin et al., 2022**; **Onorati et al., 2022**). Increased PD-L1 protein levels protect senescent cells from being cleared by cytotoxic immune cells that express the PD-1 checkpoint receptor. In fact, activation of the PD-1 receptor inhibits the cytotoxic capabilities of CD8 + T and NK cells, increasing immunosenescence. Notably, patients with MDS who possess particular somatic mutations, such as those in the TP53, ASXL1, SETBP1, TET2, SRSF2, and RUNX1 genes, have an increased propensity to react favourably to PD-1/PD-L1 inhibitors (**Chien et al., 2021**) confirming that many cellular and molecular mechanisms, known to promote cellular senescence, including alteration of the splicing machinery, are crucial stimulators of the expression of the PD-L1 protein. Interestingly, in our analysis, we also observed a correlation between the senescence gene signature score and the expression of the PD-L1 gene in CD34 + cells (**Figure 6—figure supplement 1**), supporting the previous findings linking PD-L1 gene expression to cellular senescence.

The immunology and ageing features extracted from the MDS transcriptomic data used in our analysis pipeline can enhance the conventional risk-scoring systems for MDS by providing new insights into this disease, particularly in the context of inflammation and ageing. For some patients, the clinical and genetic features may remain relatively the same until follow-up. Still, the transcriptomic features might differ considerably from the baseline diagnosis, affecting the course of treatment.

This study contributed to a deeper understanding of MDS pathogenesis and identified potential prognostic markers for this disease. It also elucidated the importance of considering the relationships between different pathways, markers, and mutations in predicting patient outcomes, highlighting the efficacy of a comprehensive approach that goes beyond all the scoring systems that have been described thus far for MDS.

## Materials and methods
### Generating gene signature scores using *singscore*
For the immunology and inflammation/aging gene sets, we performed a meticulous literature review and generated a list of gene sets from previously published articles (*Supplementary file 3*). The markers for the cellular composition gene sets were taken from *van Galen et al., 2019*. For each of the curated gene sets, instead of looking at individual gene expressions, we used *singscore* (version 1.20.0) (*Foroutan et al., 2018*), a method that scores gene signatures in a cohort of samples using rank-based statistics on their gene expression profiles. For RNA-seq data, we provided reads per million (RPM) normalized expression values to *singscore*. In the case of microarray data, we first compiled the microarray expression matrix for average expression values for all probes overlapping each gene using the *limma* package (*Ritchie et al., 2015*) in R; then we used the rankGenes function from the *singscore* package to rank each gene sample-wise. Eventually, the multiScore function was used to calculate signature scores for all gene sets at once.

### MOFA
MOFA was applied on seven views derived from the BMMNC and CD34 + cohorts: the immune profile, cell-type composition, inflammation/aging, genotype, RTE expression, clinical numeric, and clinical categorical views, using the *MOFA2* package (version 1.10.0) in R (*Figure 1a*). Each view consists of non-overlapping features of the same sample set of patients. Since aging and inflammation views share some gene sets, including inflammatory chemokines and cytokines, and we did not want to repeat these features through MOFA analysis, we combined these two views as the inflammation/aging views throughout this study. The views were scaled to have the same unit variance via the scale_view option from the MOFA model. The model pruned inactive factors incapable of capturing significant variance within the biological views, generating ten factors with a minimum explained variance of 2% in at least one biological view. Significant features within each view were determined based on the absolute weight threshold above 0.5 in at least one of the ten identified factors.

### Survival analysis
The association between latent factors and survival outcomes was investigated with Cox regression analysis and Kaplan-Meier curves via the R package *survival* (version 3.5–5). Within the BMMNC cohort, overall and event-free survivals were used as separate response variables in the univariate Cox regression, with latent factors employed as predictors. Additionally, in the multivariate Cox regression, age and sex were included as predictors alongside the factors. In the CD34 + cohort, only overall survival was used as the response variable for both regression analyses. Kaplan-Meier plots were constructed for factors exhibiting a significant hazard ratio in the univariate Cox regression, categorizing factor values into three groups: 'low' for the 1st quartile, 'high' for the fourth quartile, and 'intermediate' otherwise. The statistical significance between the high and low factor groups was determined using the log-rank test via the R package *survminer* (version 0.4.9).

### Differential expression and GSEA
We performed differential expression analysis in high versus low Factor 1 groups using *DESeq2* (*Love et al., 2014*) and generated p-values and statistics for each gene. The genes were sorted based on the 'stat' column in *DESeq2* and provided to the GSEA software (*Subramanian et al., 2005*) as an input.

GSEA was separately run on cancer hallmark and Reactome gene set databases. The GSEA output was the list of up or down-regulated pathways from each database.

## RTE expression

To generate the RTE expression, we mapped the RNA-seq reads to RepeatMasker to extract the reads covering the RTE regions and calculated the RPM scores for each class and family of RTEs. We included nine families from three main RTE classes: (1) CR1, L1, and L2 families from LINE; (2) Alu and MIR from SINE; and (3) ERV1, ERVL, ERVL-MaLR, and ERVK families from LTR.

## Acknowledgements

This study was supported with funding from Bristol Myers Squibb (BMS) company during this project. We thank Dr Sh. Kordasti for critical reading of the manuscript. AP and JB were supported by Blood Cancer UK (grants 13042 and 19004).

# Additional information

## Competing interests

Mohammad M Karimi: Reviewing editor, *eLife*. The other authors declare that no competing interests exist.

## Funding

| Funder | Grant reference number | Author |
| --- | --- | --- |
| Blood Cancer UK | 13042 | Andrea Pellagatti Jacqueline Boultwood |
| Blood Cancer UK | 19004 | Andrea Pellagatti Jacqueline Boultwood |

The funders had no role in study design, data collection and interpretation, or the decision to submit the work for publication.

## Author contributions

Sila Gerlevik, Warisha Mumtaz, Software, Formal analysis, Visualization, Methodology; Nogayhan Seymen, Conceptualization, Data curation, Software, Formal analysis, Visualization, Methodology, Writing - review and editing; Shan Hama, Seyed R Jalili, Writing - original draft; I Richard Thompson, Formal analysis, Visualization; Deniz E Kaya, Alfredo Iacoangeli, Methodology; Andrea Pellagatti, Jacqueline Boultwood, Resources; Giorgio Napolitani, Conceptualization, Formal analysis, Writing - original draft; Ghulam J Mufti, Conceptualization, Resources, Funding acquisition, Validation, Writing - original draft, Writing - review and editing; Mohammad M Karimi, Conceptualization, Data curation, Software, Formal analysis, Supervision, Investigation, Visualization, Methodology, Writing - original draft, Project administration, Writing - review and editing

## Author ORCIDs

Sila Gerlevik ⓘ https://orcid.org/0000-0001-6617-1310
Nogayhan Seymen ⓘ http://orcid.org/0000-0002-2379-5542
Alfredo Iacoangeli ⓘ https://orcid.org/0000-0002-5280-5017
Jacqueline Boultwood ⓘ https://orcid.org/0000-0002-4330-2928
Mohammad M Karimi ⓘ https://orcid.org/0000-0001-5017-1252

Reviewer #2 (Public Review): https://doi.org/10.7554/eLife.97096.3.sa1
Author response https://doi.org/10.7554/eLife.97096.3.sa2

# Additional files

## Supplementary files
- Supplementary file 1. Details of two RNA-seq datasets for MDS.
- Supplementary file 2. Mutation occurrence in MDS cohorts.
- Supplementary file 3. List of gene sets from previously published articles.
- Supplementary file 4. Categorical and numeric clinical features that were used in MDS cohorts.
- Supplementary file 5. Univariate and multivariate Cox regression result for patients in the BM CD34 +cohort to determine the relationship between the identified MOFA factors and OS.
- MDAR checklist

## Data availability
Scripts and data used in this study are available on Github (https://github.com/Karimi-Lab/MDS_MOFA copy archived at *Gerlevik, 2024*).

The following previously published datasets were used:

| Author(s) | Year | Dataset title | Dataset URL | Database and Identifier |
|---|---|---|---|---|
| Choudhary GS, Pellagatti A, Agianian B, Smith MA | 2022 | RNA sequencing of bone marrow CD34+ hematopoietic stem and progenitor cells from patients with myelodysplastic syndrome and healthy controls | https://www.ncbi.nlm.nih.gov/geo/query/acc.cgi?acc=GSE114922 | NCBI Gene Expression Omnibus, GSE114922 |
| Pellagatti A, Cazzola M, Giagounidis A, Perry J | 2010 | Expression data from bone marrow CD34+ cells of MDS patients and healthy controls | https://www.ncbi.nlm.nih.gov/geo/query/acc.cgi?acc=GSE19429 | NCBI Gene Expression Omnibus, GSE19429 |
| Shiozawa Y, Malcovati L, Gallì A, Pellagatti A, Karimi M, Sato-Otsubo A, Sato Y, Suzuki H, Yoshizato T, Yoshida K, Shiraishi Y, Chiba K, Makishima H, Boultwood J, Hellström-Lindberg E, Miyano S, Cazzola M, Ogawa S | 2018 | Transcriptome sequencing of myelodysplasia | https://ega-archive.org/datasets/EGAD00001003891 | European Genome-Phenome Archive, EGAD00001003891 |

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
