## [Editor Report · eLife assessment]

This manuscript uses public datasets of myelodysplastic syndrome (MDS) patients to undertake a multi-omics analysis of clinical, genomic, and transcriptomic datasets. **Useful** findings are provided by way of interesting correlations of specific mutations with inflammation and differing clinical outcomes. The evidence is **solid** and interesting, and the manuscript is of substantive value to hematologists and clinical immunologists.

---

## [Referee Report · Reviewer #2 (Public Review)]

Summary:

The authors performed a Multi-Omics Factor Analysis (MOFA) on analysis of two published MDS patient cohorts-1 from bone marrow mononuclear cells (BMMNCs) and CD34 cells (ref 17) and another from CD34+ cells (ref 15) --with three data modalities (clinical, genotype, and transcriptomics). Seven different views, including immune profile, inflammation/aging, Retrotransposon (RTE) expression, and cell- type composition, were derived from these modalities to attempt to identify the latent factors with significant impact on MDS prognosis.

SF3B1 was found to be the only mutation among 13 mutations in the BMMNC cohort that indicated a significant association with high inflammation. This trend was also observed to a lesser extent in the CD34+ cohort. The MOFA factor representing inflammation showed a good prognosis for MDS patients with high inflammation. In contrast, SRSF2 mutant cases showed a granulocyte-monocyte progenitor (GMP) pattern and high levels of senescence, immunosenescence, and malignant myeloid cells, consistent with their poor prognosis. Also, MOFA identified RTE expression as a risk factor for MDS. They proposed that this work showed the efficacy of their integrative approach to assess MDS prognostic risk that 'goes beyond all the scoring systems described thus far for MDS'.

---

## [Author Response]

The following is the authors’ response to the original reviews.

**Public Reviews:**

**Reviewer #1 (Public Review):**
In their manuscript, Gerlevik et al. performed an integrative analysis of clinical, genetic and transcriptomic data to identify MDS subgroups with distinct outcomes. The study was based on the building of an "immunoscore" and then combined with genotype and clinical data to analyze patient outcomes using multi-omics factor analysis.Strengths: Integrative analysis of RNA-seq, genotyping and clinical dataWeaknesses: Validation of the bioinformatic pipeline is incompleteMajor comments:(1) This study considered two RNA-seq data sets publicly available and generated in two distinct laboratories. Are they comparable in terms of RNA-seq technique: polyA versus rRNA depletion, paired-end sequencing, fragment length?

We want to reemphasize that the main point of this study is not to compare the BMMNC with the HSPC cohort. These datasets are not comparable because they were

collected from different cell types, and we should not expect them to be matched. We just analysed them in parallel to check how much HSPCs contribute to the molecular signatures we see in BMMNC samples. However, we agree with the reviewer that similar RNA-seq experimental techniques should be employed to control for confounding factors. Here is the information that we found for HSPC and BMMNC RNA-seq studies:

HSPC RNA-seq cohort: Total RNA was extracted using TRIzol (Thermo Scientific), and Sequencing was performed on an Illumina HiSeq4000 with 100-bp paired-end reads.

BMMNC RNA-seq cohort: The RNA was extracted with TRIzol reagent (Thermo Scientific). RNA-sequencing libraries were prepared from poly(A)-selected RNA and were sequenced using Illumina HiSeq 2000 or 2500 platform with 100-bp paired-end reads.

The only difference between the two cohorts is that one cohort includes total RNAs, whereas the other has polyA-selected RNAs. Since the gene set signatures use the expression of proteincoding genes, which all have polyA tails and are included in total RNA libraries, the analysis will not be affected by total vs. polyA-selected RNA-seq techniques.

(2) Data quality control (figure 1): the authors must show in a graph whether the features (dimensions) of factor 1 were available for each BMMNC and CD34+ samples.

By features of Factor 1, we think the reviewer means the features with high weights for Factor 1 in BMMNC and CD34+ samples. Figure 2c-d clearly illustrates the important features and their associations with Factor 1 for all samples in both cohorts. The samples are the columns of the two heatmaps.

(3) How to validate the importance of "immunoscore"? If GSEA of RNA-seq data was performed in the entire cohort, in the SF3B1-mutated samples or SRSF2-mutated samples (instead of patients having a high versus low level of factor 1 shown in Sup Fig. 4), what would be the ranking of Hallmarks or Reactome inflammatory terms among the others?

Our GSEA analysis was an attempt to validate the importance of our identified factors. As described in the paper, Factor 1 represents a combination of immunology scores (or “immunoscores”) in CD34+ cohort. Applying GSEA, we identified upregulation of inflammation related pathways, chemokines, and Neutrophils in patients having high (4th quartile) versus low (1st quartile) levels of Factor 1. Interestingly, sorting patients by Factor 1 resulted in similar pattern based on gene signature scores (Figure 2d).

To show that Factor1 generated by MOFA is important and different from known MDS categories such as SF3B1 and SRSF2 mutants, we performed GSEA in SF3B1-mutated vs. SF3B1-WT samples and SRSF2-mutated vs. SRSF2-WT samples in the CD34+ cohort. As shown in Author response image 1, we did not see the upregulation of inflammation and interferon pathways in SF3B1 and SRSF2 mutant MDS.

**Author response image 1. sa2fig1:** GSEA showed no upregulation of inflammation and interferon pathways for SF3B1 and SRSF2 mutant in CD34+ cohort.

(4) To decipher cell-type composition of BMMNC and CD34+ samples, the authors used van Galen's data (2019; supplementary table 3). Cell composition is expressed as the proportion of each cell population among the others. Surprisingly, the authors found that the promonocytelike score was increased in SF3B1-mutated samples and not in SRSF2-mutated samples, which are frequently co-mutated with TET2 and associated with a CMML-like phenotype. Is there a risk of bias if bone marrow subpopulations such as megakaryocytic-erythroid progenitors or early erythroid precursors are not considered?

We thank the reviewer for their insightful comment about CMML and the high prevalence of SRSF2 mutation (> 45%) in CMML cases. Using single-cell RNA sequencing and high-parameter flow cytometry, Ferrall-Fairbanks et al. (DOI: 10.1158/2643-3230.BCD-21-0217) recently showed that CMML can be classified into three differentiation trajectories: monocytic, megakaryocyte-erythroid progenitor (MEP), and normal-like. One hallmark of monocytic-biased trajectory was the enrichment of inflammatory granulocyte–macrophage progenitor (GMP)-like cells, which we observed through our analysis for SRSF2 mutants (Figure 6a).

Unfortunately, van Galen's data does not provide any gene set for MEP, and there is no singlecell RNA-seq atlas for MDS to employ to calculate the MEP score. Also, we compared the Promono-like and GMP-like gene sets from van Galen's data, and we could not find any overlap, meaning that Promono-like is not specific enough to capture the signatures coming from the more differentiated progenitors such as GMPs. Therefore, as described in the paper, we focused on GMP-like rather than Promono-like.

(5) Figures 2a and 2b indicated that the nature of retrotransposons identified in BMMNC and CD34+ was dicerent. ERVs were not detected in CD34+ cells. Are ERVs not reactivated in CD34+ cells? Is there a bias in the sequencing or bioinformatic method?

As described above, the two cohorts' sequencing methods, read length, etc., are identical.

CD34+ RNA-seq is total RNA-seq that includes both polyA and non-polyA RTE transcripts.

Therefore, the chance of bias and missing RTE signatures in CD34+ cohort is very low. L1 and Alu, which are shared between the two cohorts, are the two RTE families that are still active and make new insertions in humans. Our interpretation is that ERV activation in BM is associated with immune cells. As shown by Au et al. (DOI: 10.1016/j.ccell.2021.10.001), several ERV loci had expression in purified immune cell subsets in renal cell carcinoma samples, potentially explaining ERV upregulation in tumours responding to treatment as those biopsies had increased tumour infiltration.

(6) What is the impact of factor 1 on survival? Is it dicerent between BMMNC and CD34+ cells considering the distinct composition of factor 1 in CD34+ and BMMNC?

As shown in Table 1, Factor 1 in the BMMNC cohort is associated with overall survival (P-val < 0.05) when we did multivariate analysis but not univariate analysis. We did not observe any association between Factor 1 and event-free survival in the BMMNC cohort. Also, The 10 factors identified by MOFA in BM CD34+ cohort did not show any significance associated with MDS overall survival (Supplementary Table 5).

(7) In Figure 1e, genotype contributed to the variance of in the CD34+ cell analyses more importantly than in the BMMNC. Because the patients are dicerent in the two cohorts, dicerences in the variance could be explained either by a greater variability of the type of mutations in CD34 or an increased frequency of poor prognosis mutations in CD34+ compared to BMMNC. The genotyping data must be shown.

The genotype has already been reported in Supplementary Table 2. In fact, the number of inspected genes was much higher in the BMMNC cohort (17 genes) compared to the CD34+ cohort (3 genes). Therefore, we have more significant variability of the type of mutations in the BMMNC cohort compared to the CD34+ cohort. For the CD34+ cohort, we only had mutations for three spliceosome genes, where most cases (n=28) were SF3B1 mutants with good prognosis. We think that the result makes sense because the less genetic variability, the more homogenous groups and the more chance that one factor or a group of factors can explain the genetic variance.

(8) Fig. 2a-b: Features with high weight are shown for each factor. For factor 9, features seemed to have a low weight (Fig. 1b and 1c). However, factor 9 was predictive of EFS and OS in the BMMNC cohort. What are the features driving the prognostic value of factor 9?

As shown in Figure 3b, The main features are RTE expression from LTR:ERV1, SINE:MIR, and SINE:Alu family.

(9) The authors also provided microarray analyses of CD34+ cell. It could be interesting to test more broadly the correlation between features identified by RNA-seq or microarrays.

The microarray data did not come with any genetic information or clinical data except survival information. Therefore, we could not apply MOFA on Microarray data. However, we did generate gene signature scores from Microarray data and investigated the relationship between inflammatory chemokines and cytokines, and IFN-I signature scores with MDS survival (Figure 3c and 4c).

(10) The authors should discuss the relevance of immunosenescence features in the context of SRSF2 mutation and extend the discussion to the interest of their pipeline for patient diagnosis and follow up under treatments.

We have added the below text to the discussion:

Recent studies have shown that the expression of programmed death-ligand 1 (PD-L1) protein is significantly elevated in senescent cells (DOIs: 10.1128/mcb.00171-22, 10.1172/JCI156250, 10.1038/s41586-022-05388-4). Increased PD-L1 protein levels protect senescent cells from being cleared by cytotoxic immune cells that express the PD-1 checkpoint receptor. In fact, activation of the PD-1 receptor inhibits the cytotoxic capabilities of CD8 + T and NK cells, increasing immunosenescence.

Notably, patients with MDS who possess particular somatic mutations, such as those in the TP53, ASXL1, SETBP1, TET2, SRSF2, and RUNX1 genes, have an increased propensity to react favourably to PD-1/PD-L1 inhibitors (DOIs: 10.1111/bjh.17689, https://doi.org/10.1182/blood2020-141100) confirming that many cellular and molecular mechanisms, known to promote cellular senescence, including alteration of splicing machinery, are crucial stimulators of the expression of PD-L1 protein. Interestingly, in our analysis, we also observed a correlation between the senescence gene signature score and the expression of the PD-L1 gene in CD34+ cells (Supplementary Figure 7), supporting the previous findings linking PD-L1 gene expression to cellular senescence.

The immunology and ageing features extracted from the MDS transcriptomic data used in our analysis pipeline can enhance the conventional risk-scoring systems for MDS by providing new insights into this disease, particularly in the context of inflammation and ageing. For some patients, the clinical and genetic features may remain relatively the same until follow-up. Still, the transcriptomic features might differ considerably from the baseline diagnosis, affecting the course of treatment.

**Reviewer #2 (Public Review):**
The authors performed a Multi-Omics Factor Analysis (MOFA) on analysis of two published MDS patient cohorts-1 from bone marrow mononuclear cells (BMMNCs) and CD34 cells (ref 17) and another from CD34+ cells (ref 15) --with three data modalities (clinical, genotype, and transcriptomics). Seven different views, including immune profile, inflammation/aging, Retrotransposon (RTE) expression, and cell-type composition, were derived from these modalities to attempt to identify the latent factors with significant impact on MDS prognosis.SF3B1 was found to be the only mutation among 13 mutations in the BMMNC cohort that indicated a significant association with high inflammation. This trend was also observed to a lesser extent in the CD34+ cohort. The MOFA factor representing inflammation showed a good prognosis for MDS patients with high inflammation. In contrast, SRSF2 mutant cases showed a granulocyte-monocyte progenitor (GMP) pattern and high levels of senescence, immunosenescence, and malignant myeloid cells, consistent with their poor prognosis. Also, MOFA identified RTE expression as a risk factor for MDS. They proposed that this work showed the efficacy of their integrative approach to assess MDS prognostic risk that 'goes beyond all the scoring systems described thus far for MDS'.Several issues need clarification and response:(1) The authors do not provide adequate known clinical and molecular information which demonstrates prognostic risk of their sample cohorts in order to determine whether their data and approach 'goes 'beyond all the scoring systems described thus far for MDS'. For example, what data have the authors that their features provide prognostic data independent of the prior known factors related to prognosis (eg, marrow blasts, mutational, cytogenetic features, ring sideroblasts, IPSS-R, IPSS-M, MDA-SS)?

We agree with the reviewer that we did not generate a new cumulative risk score and compare it with the conventional risk scores for MDS. However, we identified individual MOFA factors, which are risk or protective factors for MDS, based on survival analysis in the BMMNC cohort. One reason that we did not generate our independent, cumulative score and compare it with other scores was that we did not receive any conventional risk score for the BMMNC cohort. However, we had access to all the clinical and genetic variables from the BMMNC cohort (except for three patients) that were required to calculate IPSS-R; hence, we calculated the IPSS-R in our resubmission for the BMMNC cohort. We made three IPSS-R risk categories by combining low and very low as low risk, and high and very high as high risk, and keeping intermediate as intermediate risk. Our survival analysis of these three categories showed a clear match between IPSS-R score and MDS survival (Author response image 2a).

We then investigated the relationship between factors 2, 4, and 9 from MOFA with three IPSS-R risk groups. Integration of IPSS-R risk groups with factor values confirmed the finding in the manuscript that Factors 4 and 9 generally exert a protective influence over the MDS risk, whilst higher levels of Factor 2 predict a high-risk MDS (Author response image 2b). However, we see so many outliers in all three factors, indicating that some patients were assigned to the wrong IPSS-R categories because IPSS-R calculation is based on clinical and genetic variables and does not include the transcriptomics data for coding and non-coding genomic regions.

**Author response image 2. sa2fig2:** Comparison of IPSS-R risk categories and MOFA risk and protective factors.

(2) A major issue in analyzing this paper relates to the specific patient composition from whom the samples and data were obtained. The cells from the Shiozawa paper (ref 17) is comprised of a substantial number of CMML patients. Thus, what evidence have the authors that much of the data from the BMMNCs from these patients and mutant SRSF2 related predominantly to their monocytic dicerentiation state?

We thank the reviewer for the insightful comment about the monocytic differentiation state of CMML and SRSF2 mutant cases. The BMMNC cohort has 11 CMML and 17 SRSF2 mutant cases, of which six are shared between the two groups. We have divided the patients into four groups: CMML only, SRSF2 mutant only, CCML and SRSF2 mutant, and others. We have generated boxplots for all cellular composition gene signature scores for these groups and compared the scores between these groups. As explained above, Ferrall-Fairbanks et al. (DOI: 10.1158/2643-3230.BCD-21-0217) recently showed that CMML can be classified into three differentiation trajectories: monocytic, megakaryocyte-erythroid progenitor (MEP), and normal-like. One hallmark of monocytic-biased trajectory was the enrichment of inflammatory granulocyte–macrophage progenitor (GMP)-like cells, which we observed through our analysis for the CMML cases with SRSF2 mutation (Author response image 3.).

**Author response image 3. sa2fig3:** Cellular composition gene signature scores for CMML and SRSF2 mutant versus other cases. CMML cases with SRSF2 mutation show a significant higher level of GMP and GMP-like scores compared to other MDS cases.

(3) In addition, as the majority of patients in the Shiozawa paper have ring sideroblasts (n=59), thus potentially skewing the data toward consideration mainly of these patients, for whom better outcomes are well known.

We disagree with the reviewer. We used 94 BMMNC samples from Shiozawa’s paper, of which 19 cases had Refractory Anemia with Ring Sideroblasts (RARS), 4 cases had Refractory Anemia with Ring Sideroblasts and thrombocytosis (RARS-T), and 5 cases had Refractory cytopenia with multilineage dysplasia and ring sideroblasts (RCMD-RS). In total, we had 28 cases (~30%) with Ring Sideroblasts (RS), which are not large enough to skew the data.

(4) Further, regarding this patient subset, what evidence have the authors that the importance of the SF3B1 mutation was merely related to the preponderance of sideroblastic patients from whom the samples were analyzed?

We had 34 SF3B1 mutant cases, of which 25 had Ring Sideroblasts (RS). The total number of cases with RS in the BMMNC cohort was 28. Therefore, the BMMNC cohort is not an RSdominant cohort, and RS cases did not include all SF3B1 mutants. Furthermore, it was recently shown by Ochi et al. (DOI: 10.1038/s41598-022-18921-2) that RS is a consequence of SF3B1K700E mutation, and it is not a cause to affect the SF3B1 importance.

(5) An Erratum was reported for the Shiozawa paper (Shiozawa Y, Malcovati L, Gallì A, et al. Gene expression and risk of leukemic transformation in myelodysplasia. Blood. 2018 Aug 23;132(8):869-875. doi: 10.1182/blood-2018-07-863134) that resulted from a coding error in the construction of the logistic regression model for subgroup prediction based on the gene expression profiles of BMMNCs. This coding error was identified after the publication of the article. The authors should indicate the ecect this error may have had on the data they now report.

Thank you for bringing this important issue to our attention. The error resulted from a mistake in the construction of the logistic regression model for subgroup prediction based on the gene expression profiles of BMMNCs. However, this issue does not affect our result because we analysed the expression data from scratch and generated our own gene signature scores. Also, the error has no impact on the genetics and clinical information that we received from the authors.

(6) What information have the authors as to whether the dicering RTE findings were not predominantly related to the dicerentiation state of the cell population analyzed (ie higher in BM MNCs vs CD34, Fig 1)? What control data have the authors regarding these values from normal (non-malignant) cell populations?

As described above, L1 and Alu, the two RTE families shared between the two cohorts, are still active and make new insertions in humans (Figure 2.a-b). Our interpretation is that ERV activation in BM is associated with immune cells. This interpretation is further supported by the findings of Au et al. (DOI: 10.1016/j.ccell.2021.10.001), where several ERV loci had expression in purified immune cell subsets in renal cell carcinoma samples.

Unfortunately, none of these two cohorts had normal (non-malignant) cell populations. We think that the MOFA unbiased way of modelling the heterogeneity is su@icient to capture the RTE derepressed phenotype of a subset of MDS cases compared to others, and we do not need normal cases to further support the finding.

(7) The statement in the Discussion regarding the ecects of SRSF2 mutation is speculative and should be avoided. Many other somatic gene mutations have known stronger ecects on prognosis for MDS.

One aim of this study is to identify specific immune signatures associated with SRSF2 and SF3B1 mutations, which are highly prevalent in MDS. Although other mutations, such as TP53, may have a stronger correlation with poor survival, numerous studies have demonstrated a clear link between SRSF2 mutations and poor prognosis.